

# Process mining applications in healthcare: a systematic literature review

Lerina Aversano[1], Martina Iammarino[2], Antonella Madau[3], Giuseppe Pirlo[4] and Gianfranco Semeraro[5]

[1] Department of Agricultural Science, Food, Natural Resources and Engineering, University of Foggia, Foggia, Italy
[2] Department of Information Science and Technology, Pegaso University, Naples, Italy
[3] Department of Engineering, University of Sannio, Benevento, Italy
[4] Department of Computer Science, University of Bari Aldo Moro, Bari, Italy
[5] Department of Science Technology and Society, University School for Advanced Studies IUSS Pavia, Pavia, Italy

## ABSTRACT

Process mining applications in healthcare is a field widely investigated in the last years. Its diffusion is driven by increasing digitalization and the availability of large quantities of clinical data, enabling hospitals, clinics, and other healthcare organizations to optimize workflows, reduce operational costs, and improve asset management. The importance of process mining lies in its potential to identify inefficiencies in processes, standardize clinical practices, support evidence-based decisions and, in general, improve the quality of care provided. The article aims to systematically review the research landscape in the field of process mining in healthcare, providing an in-depth understanding of how process mining is applied in healthcare. It contributes to the existing literature by highlighting the following aspects: the specific research topics covered (i), the extent of use of various process mining algorithms in different healthcare applications, showing their adaptability and effectiveness in specific contexts (ii), and, finally, the types and characteristics of data employed in these studies, highlighting the needs and challenges related to data in healthcare process mining (iii). Through this systematic literature review, the article can support researchers in identifying the most valuable research topic to be explored by the scientific community working on process mining in healthcare. To achieve this goal, several articles focusing on the algorithms and data employed were selected and analyzed. The final discussion highlights current research gaps, suggesting future areas of investigation, and identifies critical issues and vulnerabilities of existing process mining applications in healthcare.

## INTRODUCTION

Process mining is a research area that provides innovative models to analyze and optimize business processes.

Compared to traditional business process management, which relies on meetings, interviews, and direct observations, and thus deals with a limited view of processes, process mining provides a more comprehensive approach to examining, improving, and

Corresponding author
Martina Iammarino,
martina.iammarino@unipegaso.it

monitoring business processes. Therefore, by using event data and innovative algorithms, process mining can effectively support business process optimization (*van der Aalst, 2012b*).

An event log consists of a chronological sequence of events that trace the execution of business process activities. Process mining uses event log data to address process-related issues. An event log adopts the notion of a case to correlate events. An event log consists of events that generally refer to a case, that is, a particular process instance, an activity, and a timestamp. In addition, there may be additional event attributes that refer to resources, people, costs, *etc*. Leveraging an event log is a valuable means to analyze and improve business processes (*Dakic et al., 2020*).

In particular, process mining is successfully used in various fields, including healthcare, where it has led to significant improvements in care quality, resource optimization, and patient satisfaction (*Munoz-Gama et al., 2022*; *Erdoğan & Tarhan, 2016*). The healthcare domain is characterized by the complexity of the processes and related managed data. They significantly change over time due to various factors, including patient conditions and different methods of task execution (*Tamburis & Esposito, 2020*).

Specifically, healthcare processes involve a series of activities aimed at diagnosing, treating, and preventing diseases to improve the patient's health and to monitor the patient's journey from admission to discharge. These processes are supported by clinical and non-clinical activities carried out by various resources such as medical doctors, nurses, technicians, and clerks, and can vary from one organization to another. Analyzing and improving these processes is complex due to their dynamic, personalized, and increasingly multidisciplinary nature. However, achieving this goal is challenging and involves numerous obstacles, including the need to reduce costs, improve demand management, decrease waiting times, increase resource productivity, and enhance process transparency.

Several studies highlight the benefits of process mining in healthcare with real-world case studies, obtaining significant improvement in operational efficiency and patient care. In *Arias et al. (2020)*, for example, the authors highlighted how process mining techniques were used to analyze patient journeys in a healthcare facility, enabling the identification of bottlenecks and optimization of patient flow, ultimately leading to improved experiences and outcomes. Similarly, *Hendricks (2019)* reports on a study utilizing process mining tools to analyze the event logs of patients with sepsis In emergency room management. This approach allowed researchers to identify inefficiencies in patient flows, leading to reduced waiting times and enhanced care quality while uncovering insights that traditional observation methods may have missed. In gynecological oncology at the AMC (Academic Medical Center) Hospital, researchers employed process mining techniques on billing data to effectively map the complex care pathways of patients (*Mans et al., 2008*). The insights that emerged from this analysis facilitated improved resource allocation and enhanced patient outcomes through better coordination of care. *van der Aalst (2009)* proposed an accurate model to improve the ability to specify and implement process requirements in hospital information systems (HIS), supporting process analysis, and allowing the automatic construction of simulation models.

Therefore, this study aims to provide a systematic literature review identifying and analyzing applications of process mining algorithms in the healthcare domain. The study reports an overview on the state of the art of the research studies in the field, to support researchers on the use of process mining techniques, and emphasize the benefits of integrating this discipline.

The rest of the document is structured as follows: "Related Work" reports the related works and in "Background" discusses the context relating to the research topic and its current development. "Research Method" outlines the search methodology used by providing a full explanation of the search queries and selected databases, search, and filtering process. "Results" reports the results of our research work based on the defined research questions, "Discussion" interprets and analyzes the results obtained, relating them to the broader context of process mining in healthcare, and finally "Conclusions" summarizes the findings of the research work.

## RELATED WORK

In recent years, process mining has increasingly drawn the attention of the healthcare sector as a valuable tool for analyzing and optimizing workflows. Several research studies have thoroughly assessed the current state of the industry, providing an up-to-date overview of its applications and benefits. Reviews on process mining in healthcare can generally be classified into two main groups. The first group includes generic reviews that analyze the application of process mining in healthcare in general, exploring its benefits and potential in various contexts, such as most of those present in Table 1. The second group consists of reviews that focus on a particular healthcare subdomain, such as treatments for specific diseases or specific medical conditions. These reviews offer a more detailed view of the challenges and opportunities of process mining in a specific context. In *Grüger et al. (2020)* and *Kurniati et al. (2016)*, the specific topic is the field of oncology while in *Kusuma, Hall & Johnson (2018)* the focus is on how process mining is used in the healthcare sector for cardiac diseases, analyzing several works. During the exploration carried out to understand the fundamental aspects of a mining process review, *Chen et al. (2023)* include only types of chronic diseases, such as diabetes or obesity, and shows how the process mining is also present in more specific fields. Upon analysis, all the reviews highlight the significant potential of process mining as a valuable tool for improving the efficiency, effectiveness, and quality of healthcare.

Overall, by extracting information from event logs, one can identify bottlenecks, inefficiencies, and even compliance issues. The processes can then be redesigned to reduce costs, and consumption, and improve the conditions of the healthcare service offered.

In Table 1, there are 11 review articles on process mining in the healthcare sector published since 2014. Only *Partington et al. (2015)* is geographically localized and presents an analysis of processes present in four Australian hospitals.

On the other hand, regarding the concept of deep learning, which has been growing in recent years (*Alzoubi, Mishra & Topcu, 2024*; *Aggarwal et al., 2022*), only one work (*Chen et al., 2023*) is taken into consideration for exploration works that concern this part.

**Table 1  Previous literature reviews.**

| Reference | Title | Publication venue | Year | Syst. review | Health focused | DL focused | Datasets description |
|---|---|---|---|---|---|---|---|
| *Partington et al. (2015)* | Process mining for clinical processes: A comparative analysis of four australian hospitals | ACM Transactions on Management Information Systems | 2015 | No | Yes (results from case studies) | No | No |
| *Rojas et al. (2016)* | Process mining in healthcare: A literature review | Journal of Biomedical Informatics | 2016 | No | No | No | Almost-data type description |
| *Erdogan & Tarhan (2018)* | Systematic Mapping of Process Mining Studies in Healthcare | IEEE Access | 2018 | Yes | Almost | No | No |
| *Iglesias, Juarez & Campos (2020)* | Comprehensive analysis of rule formalisms to represent clinical guidelines: Selection criteria and case study on antibiotic clinical guidelines | Artificial Intelligence in Medicine | 2020 | No | Yes | No | No |
| *Dallagassa et al. (2022)* | Opportunities and challenges for applying process mining in healthcare: a systematic mapping study | Journal of Ambient Intelligence and Humanized Computing | 2022 | Yes | Almost | No | No |
| *Guzzo, Rullo & Vocaturo (2022)* | Process mining applications in the healthcare domain: A comprehensive review | Wiley Interdisciplinary Reviews: Data Mining and Knowledge Discovery | 2022 | Yes | Yes | No | Almost-some type of datasets are described |
| *De Roock & Martin (2022)* | Process mining in healthcare–An updated perspective on the state of the art | Journal of Biomedical Informatics | 2022 | Yes | No | No | No |
| *Oliart, Rojas & Capurro (2022)* | Are we ready for conformance checking in healthcare? Measuring adherence to clinical guidelines: A scoping systematic literature review | Journal of Biomedical Informatics | 2022 | Yes | Yes | No | Almost-the characteristics of data are described |
| *Manktelow et al. (2022)* | Clinical and operational insights from data-driven care pathway mapping: a systematic review | BMC Medical Informatics and Decision Making | 2022 | Yes | Yes | No | No |
| *Chen et al. (2023)* | Process mining and data mining applications in the domain of chronic diseases: A systematic review | Artificial Intelligence in Medicine | 2023 | Yes | Yes | Almost-just two articles with deep learning are cited | No |

To fully understand the current state of the art and compare it with our contribution, it is important to note that some works (*Rojas et al., 2016*; *Guzzo, Rullo & Vocaturo, 2022*; *De Roock & Martin, 2022*) offer a partial and non-exhaustive description of the datasets used in the field of process mining in healthcare but without focusing on the characteristics of the datasets.

As regards the areas of greatest interest related to process mining, there are three groups of techniques: process discovery which generates a process model from the event log without *a priori* model, conformance checking which compares a process model with its

event log where the inputs are the event log and the process model, while the output is diagnostic information showing differences and commonalities between the model and the log, and process enhancement which takes an existing process model and attempts to extend or improve the model by adding more data or using observed events. *Augusto et al. (2019)* in concentrate on the field of process discovery. Their article offers a comprehensive evaluation comparing various automated process discovery methods and this evaluation leverages an open-source benchmark and analyzes twelve publicly available real-world event logs, along with twelve private logs from process owners considering nine quality metrics to evaluate the methods' performance. Several studies (*Oliart, Rojas & Capurro, 2022*; *Dunzer et al., 2019*; *Saini, Kamra & Shrivastava, 2021*) have evaluated the effectiveness of various approaches in conformance checking. Finally, *Yasmin, Bukhsh & De Alencar Silva (2018a)* contribute to the field of process optimization by providing a comprehensive review of 43 relevant articles.

# BACKGROUND

The contemporary context of organizations is characterized by a growing dependence on information systems and the efficient management of business processes. In this context, process mining emerges as an important discipline that combines principles of data mining, process analysis, and information management to derive value from companies' operational data. This methodology focuses on extracting, analyzing, and visualizing process data to understand how activities within an organization take place. The evolution of process mining has been catalyzed by the need to improve operational efficiency, optimize workflows, and ensure regulatory compliance.

This section of the document explores the theoretical and practical context of process mining, providing a detailed overview of the conceptual foundations of process mining.

## Introduction to process mining

Process mining is an advanced data analysis methodology that allows you to discover, monitor, and improve business processes by extracting knowledge from event logs recorded in information systems (*van der Aalst, 2012a*). It arises from the need to bridge the gap between the theoretical model of the processes and their real execution. Traditionally, companies have relied on static models to describe their processes, but these models often do not accurately reflect the complexity and variability of daily operations. Process mining offers a dynamic, data-driven approach that can provide an accurate and detailed representation of how processes work.

Event logs are the primary source of data used in process mining. These logs are detailed records of activities performed within company information systems. A well-structured event log contains several key elements that allow you to accurately track operations and analyze business processes. These logs include key information such as the timestamp, which indicates when an activity was executed, the case identifier (case ID), which represents a single instance of the process, and the activity type, which describes the action performed (*van der Aalst, 2011*). In addition to the main fields, an event log can contain other additional attributes that provide additional details about the context of the activity.

These may include the type of product, the cost associated with the activity, notes or comments, and any other relevant information that can help in analyzing the process. The collection and analysis of this data requires a preliminary pre-processing phase, where the data is cleaned and structured to ensure its quality and consistency. This process may include removing duplicates, handling missing values, and aligning information to ensure the data is ready for analysis. Once pre-processed, the data is analyzed using specific process mining algorithms, which allow accurate and detailed process models to be built (*Marin-Castro & Tello-Leal, 2021*).

Process mining stands out from other data analysis techniques for its ability to provide a visual and dynamic representation of business processes. This allows organizations to identify inefficiencies, bottlenecks, and deviations from intended processes, facilitating continuous improvement and operational optimization. Ultimately, process mining offers a data-driven approach to understanding and improving business processes, providing powerful tools for managing and optimizing operations.

Process mining is mainly divided into three types: discovery, conformance, and enhancement (*van der Aalst, 2022*). Discovery deals with the creation of process models starting from raw data, without any prior knowledge of the process itself. This allows you to visualize the real flow of activities, identifying sequences of operations and their variants. Conformance checking compares existing theoretical models with real data to verify compliance, highlighting deviations and anomalies. Enhancement, on the other hand, aims to improve process models by integrating them with detailed information extracted from operational data, thus optimizing the efficiency and quality of operations.

### Process discovery

Process discovery is a process mining technique that deals with the creation of process models from event data recorded in an organization's information systems. This approach does not require any prior knowledge of the process and is based solely on available data. The main goal of process discovery is to reveal how processes take place, offering an accurate and detailed representation of activity sequences, their interactions, and process variants (*van der Aalst, 2018*).

The initial phase of process discovery involves the collection of event logs. This data is usually extracted from enterprise information systems such as ERP, CRM, document management systems, and other process management tools. Once event logs are collected, it is often necessary to clean and pre-process them to ensure they are suitable for analysis. After preprocessing, the data is analyzed using process discovery algorithms. Among the best-known are the Alpha algorithm, the Heuristic Miner algorithm, and the Fuzzy Miner algorithm. These algorithms work to identify sequences of activities and the relationships between them, thus building a process model that represents the flow of operations as they occur. The resulting model is then visualized using process diagrams, such as flowcharts, which allow you to examine in detail the activities, execution paths, and possible variations of the process. This model can reveal valuable information, such as bottlenecks, deviations from expected procedures, and opportunities for improvement.

Finally, the process model derived from process discovery is often validated and refined through further analysis and comparisons with the expectations and knowledge of domain experts. This allows you to obtain a faithful and usable representation of the real process, which can be used to optimize operations, improve efficiency, and ensure compliance with regulations and company standards.

### Conformance checking

Conformal checking is a process mining technique that compares a predefined process model with recorded event data to verify whether real process executions adhere to the theoretical model. The main objective of conformal checking is to identify deviations between the ideal process, represented in the model, and the executed process, recorded in the event logs. This technique is critical to ensuring that business operations comply with defined policies, regulations, and standards (*Carmona et al., 2018*).

To get started, you need to have a process model that represents how the process should be executed. In parallel, you collect event logs that document the actual activities performed in the company information system.

Once both the theoretical model and the real data are ready, we proceed with the compliance analysis. The conformal checking algorithm examines the event logs and compares them to the process model to identify areas of misalignment. During this phase, the algorithm detects whether the tasks were performed in the expected order, whether all process rules were respected, and whether there were any missing or over-performed tasks.

The results of conformal checking are presented through reports and visualizations that highlight the deviations detected. These results can indicate, for example, whether there were instances where tasks were performed out of sequence, whether there were unexpected delays, or whether some tasks were skipped altogether.

Conformity checking is particularly useful in regulated contexts where process compliance is essential to ensure quality and safety. For example, in the healthcare sector, it can be used to verify that clinical protocols are followed correctly, reducing the risk of medical errors. In finance, it can help ensure operations comply with anti-money laundering regulations.

### Enhancement

Enhancement is a process mining technique that focuses on optimizing and improving existing business processes using recorded event data (*Yasmin, Bukhsh & Silva, 2018b*).

The starting point of enhancement is an existing process model, which represents the theoretical structure of the process as it should be executed. Added to this model are event logs, which document the activities carried out within the organization. By analyzing these logs, enhancement helps you identify and incorporate relevant information that can improve your process model.

One of the most common ways of enhancement is the addition of performance information. By analyzing temporal data contained in event logs, such as task start and end timestamps, you can calculate performance metrics such as cycle times, wait times, and

execution times. This information is then integrated into the process model, allowing you to identify bottlenecks, activities that take longer than expected, and areas where efficiency can be improved.

Another form of enhancement involves the addition of contextual data. This data may include additional information about the resources involved, such as the operators performing the tasks, the departments responsible, or the costs associated with each task. By integrating this data into your process model, you can gain a more complete and detailed view of your process, making it easier to identify inefficiencies and opportunities for improvement.

Enhancement can also be used to update the process model based on new findings. For example, if during the analysis of event logs new process variants are discovered that were not considered in the original model, these can be incorporated into the model to better reflect operational reality. This continuous updating process helps keep the process model aligned with actual practices and ensures that it is always relevant and useful for optimizing operations.

## Process mining in healthcare

The healthcare sector is characterized by considerable operational complexity and the need to manage a large number of critical processes to ensure quality patient care. These processes include patient flow management, resource planning, compliance with clinical protocols, and clinical data management (*Martin et al., 2020*). In this context, process mining has proven to be a valuable tool for analyzing and improving the efficiency and effectiveness of healthcare processes.

Process mining in healthcare allows you to extract knowledge from event logs generated by clinical information systems, such as electronic health record (EHR) systems, hospital management systems (HIS), and clinical decision support systems (CDSS) (*Munoz-Gama et al., 2022*). Using this information, detailed models of clinical and administrative processes can be created, revealing how they are performed and identifying any inefficiencies or deviations from standard protocols. One of the main advantages of process mining in healthcare is the ability to improve the management of patient flows. By analyzing data on patient movements across different departments, you can identify bottlenecks and delays, optimize waiting times, and improve resource allocation. For example, process mining can help reduce waiting times in emergency rooms, optimize operating room scheduling, and improve the management of hospital stays (*Rojas et al., 2019*). Another important field of application is compliance with clinical protocols (*Gatta et al., 2019*). Process mining allows you to verify whether the care provided to patients complies with established guidelines and protocols. This is crucial to ensuring the quality of care and reducing the risk of medical errors. For example, it can be used to monitor adherence to treatment protocols for chronic diseases, ensuring that patients receive the appropriate care at every stage of treatment. Process mining is also useful for performance analysis and strategic planning. By integrating performance data into process models, you can monitor key metrics such as cycle times, wait times, and costs associated with different clinical activities. This allows you to identify opportunities for improvement and make

informed decisions to optimize operations and reduce costs without compromising the quality of care.

Despite its many advantages, the application of process mining in healthcare also presents several challenges. The quality and availability of data can vary greatly between different organizations and information systems. Furthermore, managing data privacy and security is crucial, as healthcare data is highly sensitive (*Dallagassa et al., 2022*). It is therefore crucial to implement adequate measures to protect patient data during analysis.

Therefore, through this systematic review, we aim to consolidate existing knowledge, identifying the most used practices and gaps in research. This review aims to offer a balanced and comprehensive view of the potential impact of process mining in healthcare.

# RESEARCH METHOD

In this section, we illustrate in detail the methodology used to carry out the systematic review, focusing on the research questions, the databases consulted, and the filtering and search methods used. Our approach is inspired by the principles outlined by *Kitchenham (2004)*, who developed guidance for systematic reviews.

The process is divided into the following sequential phases:

- formulation of relevant research questions;
- creating search queries based on extracting keywords from search questions;
- selection of databases to consult;
- definition of the initial filtering criteria, such as the time interval and the quality of the searched results;
- screening of titles and abstracts to eliminate irrelevant articles and duplicates;
- definition of more detailed eligibility criteria to be applied during the in-depth reading of the selected articles;
- analysis of the remaining articles about the initial research questions.

These steps allowed us to conduct an accurate systematic review based on rigorous criteria.

## Research questions

Conducting a systematic review of the literature in the field of process mining in healthcare is of great importance because it allows us to collect and synthesize existing knowledge on the most relevant topics, on the approaches and algorithms used, as well as on the types of data used. This process offers a comprehensive view of the state of the art, highlighting current trends and breakthroughs. By systematically analyzing the literature, it is possible to identify areas that are still unexplored or under-studied, thus directing future research toward unresolved or little-treated questions. This promotes innovation and advancement of the field, ensuring that research efforts focus on aspects that can bring concrete benefits to the healthcare sector. Therefore, in this context, we defined three research questions to guide our study.

First of all, we considered it necessary and interesting to map the main topics covered in the existing literature to understand which aspects of process mining the attention of scholars has focused most on. In the healthcare context, this understanding is crucial as the sector is characterized by complex and multidimensional processes that require in-depth and detailed analysis to improve the efficiency and quality of care. Identifying prevalent research themes helps us identify the areas of greatest interest and impact, such as optimizing hospital workflows, reducing waiting times, improving patient management, and analyzing variability in treatment pathways. Furthermore, it allows us to detect areas that have received less attention and could benefit from further study. In this regard, we have addressed the first research question:

**RQ$_1$**: *Which research topics can be identified in the primary studies of process mining in healthcare?*

By examining the different approaches and algorithms used in the healthcare sector, it is possible to understand the quality and robustness of the available evidence. This is essential to ensure that the conclusions drawn are based on reliable data and valid methodologies, avoiding adopting practices based on weak or unconvincing evidence. Therefore, identifying and cataloging the process mining algorithms developed and used in different healthcare settings allows us to assess the breadth and variety of the techniques available, as well as highlight the distribution of the most widespread applications. Algorithms can be used to discover patterns in clinical data, identify bottlenecks in processes, and suggest interventions to optimize operations therefore analyzing the distribuiton of the proposed algorithms also allows us to understand which methodologies are most effective in certain contexts, what their limitations are, and identify areas where further development is needed. This is particularly relevant in a dynamic sector such as healthcare, where needs can vary significantly from one context to another, and where continuous innovation is necessary to meet new challenges. Therefore, it is crucial to provide a vision that connects techniques and healthcare fields, is useful for other researchers and professionals in the sector. To create a best practice framework that can be adopted by other researchers and industry professionals we address our second research question:

**RQ$_2$:** *To what extend are process mining algorithms used in each specific healthcare application?*

Finally, knowing the types of data used in process mining in healthcare allows us to understand the variety and complexity of the information available. Healthcare data can include electronic patient records, laboratory data, clinical workflows, treatment information, and other operational data. Each type of data has unique characteristics that influence how it can be analyzed and interpreted through process mining techniques. Understanding the characteristics of the data, such as their structure, granularity, quality, and provenance but also how the privacy criteria of the same are managed, the presence of complete or incomplete data is essential for selecting the most appropriate process mining algorithms and for designing studies that can provide valid and useful results. With a solid understanding of available data, researchers and practitioners can make the most of the

potential of process mining to optimize healthcare processes and obtain meaningful insights, so we address the third research question:

**RQ₃**: *What are the types of data and their characteristics?*

## Query

The use of targeted queries is crucial to identify and select the most relevant studies for the systematic review. These queries are carefully constructed, using key terms extracted from the definitive search questions, to accurately capture the literature relevant to the topic studied. The choice of terms and their combination are designed to maximize sensitivity and specificity. Querying databases with specially designed queries allows you to quickly filter through a large number of publications and focus on those that best answer your specific search questions. This approach helps reduce the risk of omitting important studies and ensures that the review is based on a comprehensive collection of relevant evidence.

Therefore, once we have identified the main research directions detailed in "Research Questions", we report below the query used to interrogate the selected databases, as detailed in the next paragraph, to identify the studies to include in the review.

**Q:** *(Process Mining) AND (Health)*

## Databases

To identify studies relevant to our systematic review, we performed extensive searches of several reference databases. These databases were selected for their breadth of coverage in the field of process mining and their ability to host a wide range of scientific publications in the healthcare sector. Using specific questioning strategies reported in "Query" and predefined inclusion criteria reported in "Methodology for Research and Document Selection", we aimed to identify studies that comprehensively answered the defined research questions. Below are the databases queried:

- IEEE Xplore: A digital library provided by the Institute of Electrical and Electronics Engineers (IEEE), one of the world's largest professional organizations for the advancement of technology. Includes journal articles, conference proceedings, technical standards, ebooks, and educational courses. Its main topics include Electrical engineering, electronics, information technology, telecommunications, robotics, energy, biomedicine, and other related areas. Continuous updates with new publications and technical documents.

- ACM Digital Library is a comprehensive collection of scientific and technical literature produced by the Association for Computing Machinery (ACM). It mainly covers the field of computer science and related disciplines such as artificial intelligence, software engineering, computer networks, cybersecurity, human-computer interaction, and much more. Includes journals, conference proceedings, technical journals, newsletters, and books published by the ACM.

- ScienceDirect: a leading academic research platform operated by Elsevier, a leading global publisher of scientific, technical, and medical literature. It hosts scientific journal

articles, book chapters, and other academic publications from Elsevier. It covers a wide range of scientific and technical disciplines, including life sciences, physical sciences, health sciences, engineering, social sciences, and humanities. It offers powerful search and navigation tools, with the ability to filter results by document type, publication date, author, and more. Additionally, it includes features for citation management and access to impact metrics.

In detail, to ensure comprehensive coverage of the relevant literature, we have used a combination of specific keywords and Boolean operators, including inserting specific keywords, such as "process mining," "healthcare optimization," and "patient flow," on each of the selected databases. We have also applied filters to insert any criteria such as time range, language, *etc.*, to maintain the relevance of the results to the healthcare field.

## Methodology for research and document selection

Having defined the research directions and the sources from which the studies were extracted for the analysis, we can now examine the search and filtering process that was followed.

The process was divided into three crucial phases, as illustrated in Fig. 1. In the first phase, we performed specific queries on the selected databases to collect relevant documents. Using the keywords of inputs composed of queries (Q) defined in "Query", the search process selected articles published in the year range of 2014 to 2024 ($IC_2$) from the databases listed in "Databases". We chose a 10-year time horizon because it offers a balance between breadth and relevance, allowing us to include a sufficient amount of studies to get a complete view without making the data obsolete. This time frame is long enough to include significant developments in the field, but not so long that current trends are difficult to interpret. This first phase of the search produced a total of 159 articles, of which 87 came from ACM, 31 from IEEE Xplore, and 41 from ScienceDirect.

Subsequently, in the second phase, we applied rigorous inclusion (IC) and exclusion (EC) criteria to filter the initially selected documents, as detailed in Table 2. We have selected articles that presented case studies, practical applications, and process mining analyses specifically in healthcare contexts. Articles that dealt exclusively with non-healthcare contexts or that provided theoretical discussions without empirical data were excluded. This approach allowed us to maintain the focus on studies that directly contribute to the analysis and optimization of healthcare processes, ensuring relevance and applicability.

In particular, we have selected only works written in English ($IC_1$) in the period 2014/2024 ($IC_2$), published in journals ($IC_3$) and which use process mining techniques in the healthcare sector ($IC_4$). We excluded from our reviews studies that were already surveys or reviews ($EC_1$) and that did not involve the use of process mining techniques ($EC_2$) or did not concern the healthcare sector ($EC_3$). Therefore, after completion of the filtering process, which involved (i) reading titles, (ii) scanning abstracts, and (iii) eliminating duplicate documents, 80 articles were found.

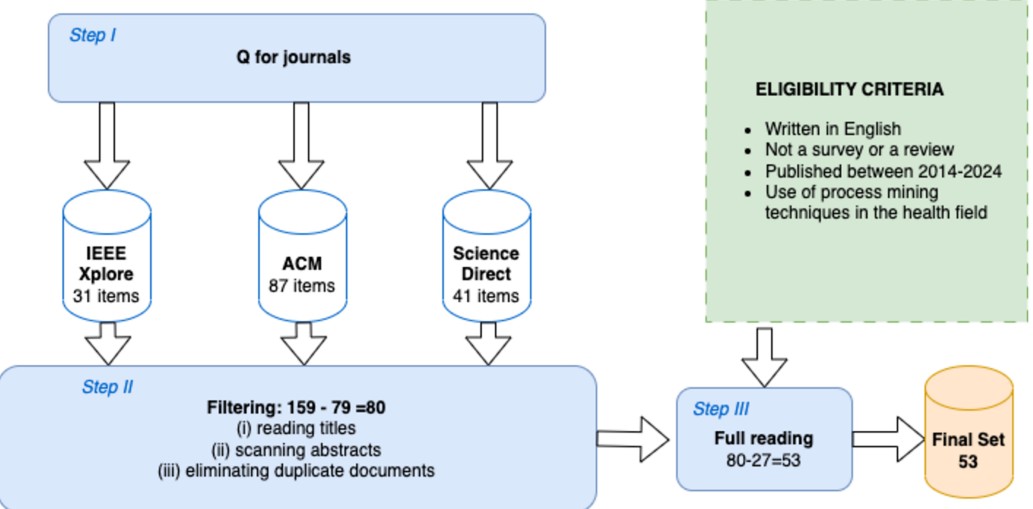

**Figure 1 Research and filtering procedure for article selection.**

**Table 2 Inclusion and exclusion criteria.**

|  | Criterium ID | Description |
|---|---|---|
| 4*Inclusion criteria | $IC_1$ | Studies written in English |
|  | $IC_2$ | Studies published in the range 2014–2024 |
|  | $IC_3$ | Studies published in a press or in a journal |
|  | $IC_4$ | Studies should use process mining techniques in the health field |
| 3*Exclusion criteria | $EC_1$ | The studies are bibliographic surveys or systematic reviews |
|  | $EC_2$ | The studies do not involve Process Mining techniques |
|  | $EC_3$ | The studies do not concern the Health field |

Finally, the third phase involved a detailed analysis of the studies included in the review, in which 53 documents were considered because they met the eligibility criteria.

For data extraction, we have used a standardized protocol that included fields such as study objective, process mining methods used, types of health data analyzed, and main results. Each reviewer independently examined the article and extracted the information and data needed to answer the research questions. The extracted data were then organized into a standardized format, with each column representing a specific characteristic identified in the individual studies. Subsequently, each reviewer's version was compared with that of the other reviewer assigned to the same article and a third review was conducted to resolve any discrepancies, thus ensuring an objective and rigorous approach. This systematic approach ensured the consistency of the extracted data and facilitated the final synthesis. We then categorized and compared the results of the studies to identify common trends, methodological differences, and potential areas for innovation.

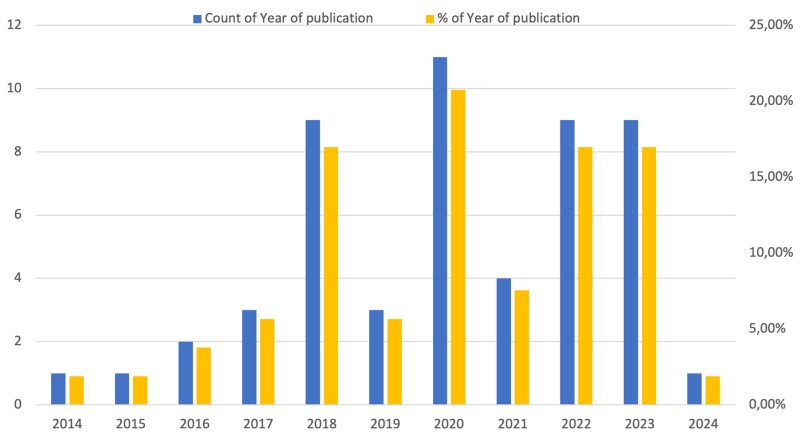

**Figure 2  Yearly analysis of article distribution.**  

Thus, in summary after the initial identification of articles, we followed a multi-stage screening process to select relevant studies. First, we eliminated duplicates, followed by screening titles and abstracts to assess compliance with the inclusion criteria. Next, we conducted a full-text review of the selected articles to confirm their relevance. Two reviewers independently reviewed the articles at each stage, with a third review to resolve any discrepancies, thus ensuring an objective and rigorous approach.

For a more detailed analysis, Fig. 2 illustrates the temporal distribution of the studies that we included in the analysis. This graph allows us to track the trends and variations in the number of studies published over the period considered. Specifically, it allows you to identify significant temporal trends, such as peaks or declines in research activity, periods of increased interest or attention on specific aspects of the topic, and any interruptions or anomalies in scientific output. By analyzing this data, researchers can better understand the evolution of the field and the challenges faced over the years. In the Figure, the years are represented on the x-axis, while the number of contributions is represented on the y-axis, indicating both the total articles in blue and the percentage in yellow. The analysis shows a significant increase in the number of relevant articles every other year between 2018 and 2021, stabilizing in 2022/2023 and decreasing in the last year taken into consideration.

We also thought it appropriate to create a graph on the distribution of articles by the publisher to understand the impact and influence of the different publishers in the field of study considered. This graph allows you to see how many publications have been produced by each publisher, thus providing an indication of editorial leadership in the sector. Furthermore, analyzing the distribution of articles by publisher can help identify publishing trends, publishing preferences among authors, and the reputation associated with each publisher in the specific research field. These data are essential for evaluating the diversity and quality of available information sources and for guiding strategic decisions regarding the dissemination and access to scientific knowledge. Therefore, Fig. 3 presents the distribution of articles included by publishers, where the names of the publishers are indicated on the x-axis and the counts are represented on the y-axis. The blue bars indicate the absolute number of articles for each publisher, while the yellow bars express the relative

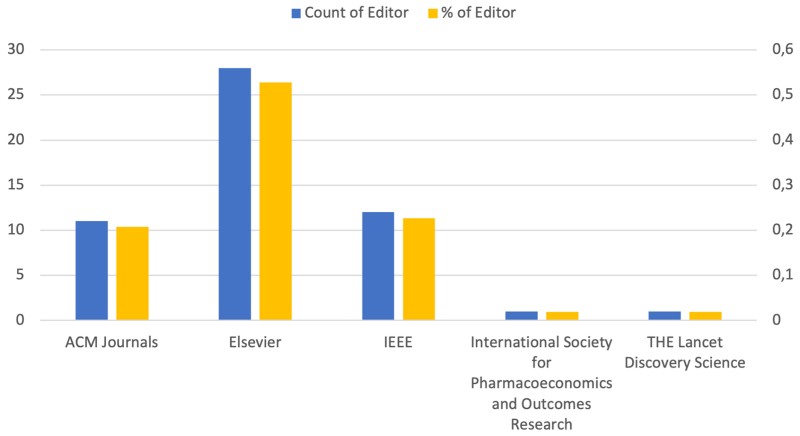

**Figure 3 Distribution of article by publisher.**

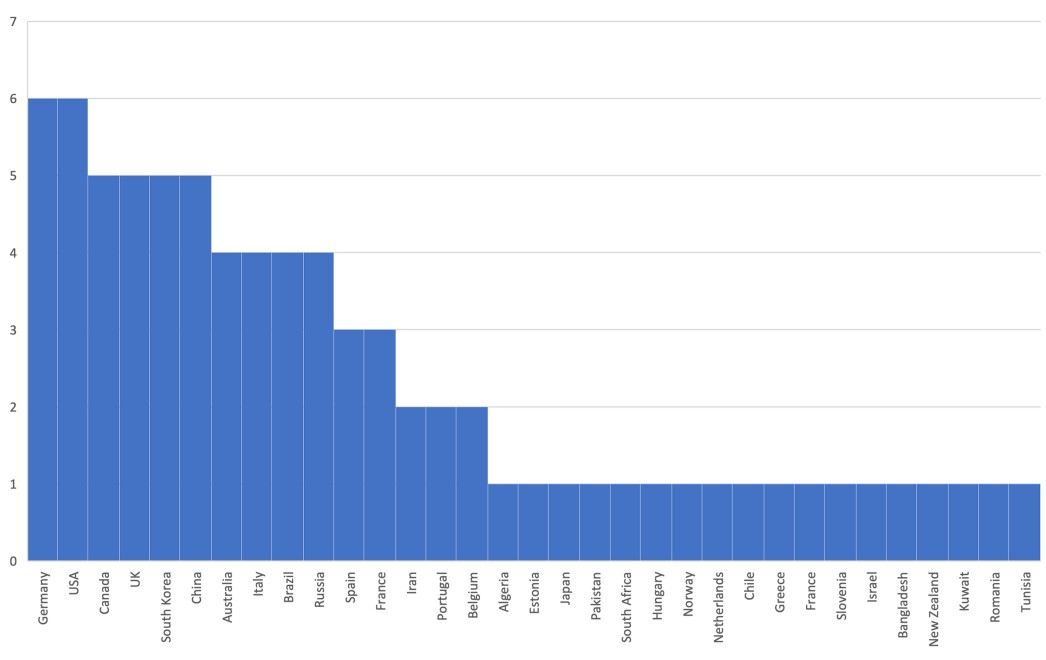

**Figure 4 Country-wise distribution of article by authors.**

percentage. As can be seen from the graph, the figure highlights the prevalence of articles published by Elsevier within our research topic, followed by ACM and IEEE in equal measure. The contribution of the International Society for Pharmacoeconomics and Outcomes Research and The Lancet Discovery Science is small.

Finally, Fig. 4 shows the distribution of the articles based on the countries of the authors involved in the study. Each bar represents a specific country, while the ordinate axis indicates the number of articles published by authors from that country. This graph is fundamental as it allows us to understand the geographical origin of the authors of the articles included in the study. Analyzing the distribution of articles by country provides insights into the geographic diversity of publications, highlighting the regions of the world

where there is greater research activity in the specific field covered. This information can be useful for identifying regional trends, comparing research activity between different countries, and evaluating the geographical impact of the topics covered in the analyzed articles. From the image, it is clear that the countries in which the most process mining studies in the healthcare sector are produced are Germany and the USA, followed by Canada, the UK, South Korea, and China.

## RESULTS

In this section, we present the findings for each research question.

### RQ1: Which research topics can be identified in the primary studies of process mining in healthcare?

To identify which process mining topic was most used in healthcare, we conducted a quantitative analysis to determine the number of documents for each topic. Specifically, we considered four areas of process mining, as defined in the (*dos Santos Garcia et al., 2019*) study: discovery, compliance, enhancement, and support areas. The last one includes articles that do not directly describe one of the three main types of process mining, but explore other related questions to the implementation of process mining projects. The results are shown in Fig. 5, where the bar graph shows the four topics considered on the ordinate axis and the values on the x-axis. From the graph, it is clear that the most explored topic is Process Discovery, with almost 30 documents, indicating significant interest in discovering current processes from data. Process enhancement and supporting areas follow, each with a more moderate number of articles, suggesting that process improvement and support areas are also relevant but less central topics than process discovery. Finally, conformance checking is the least covered topic, with less than 10 articles, which could indicate that verifying the conformity of existing processes concerning expected standards or models is a field still under development or less of a priority than the other topics.

Figure 6 illustrates the annual distribution of articles by topic from 2014 to 2024 using a stacked histogram. The various colors represent different categories of topics covered: orange for support areas, gray for process improvement, red for process discovery, and blue for compliance verification. From 2014 to 2016, the number of articles published was very low, with one or two articles per year, mostly in support areas. In 2017, an increase in the number of publications was observed, with a diversification of the topics covered, including process improvement and process discovery. The year 2019 represents a significant peak with a large number of articles, especially in the areas of support and process improvement, as process discovery and compliance verification begin to emerge as topics of interest. In 2020, we reached an all-time high in publications, with strong growth in all topics, especially in the areas of support and process improvement. The years 2022 and 2023 show a recovery, with a significant number of articles published, again with a good representation of all topics, and a slight predominance in support and process discovery. In summary, the figure highlights a growth trend in publishing articles on

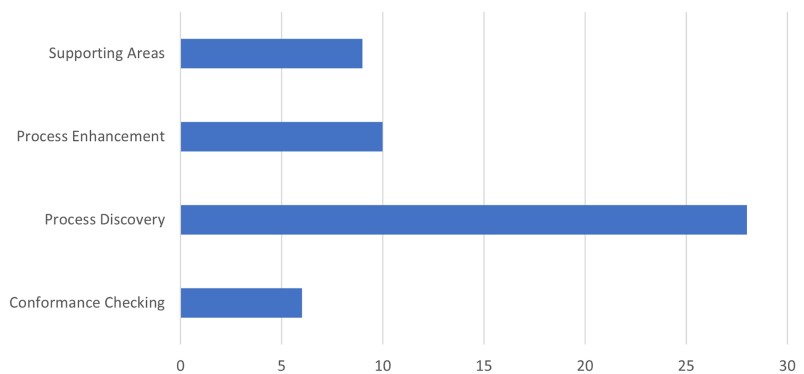

**Figure 5  Number of article for each process mining topic applied in healthcare.**

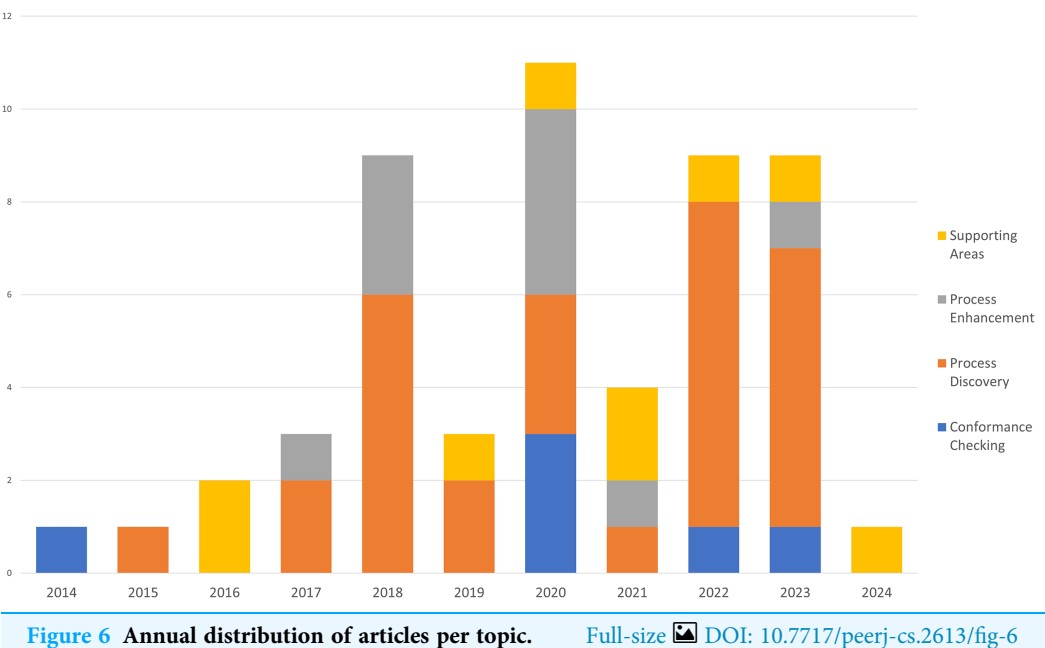

**Figure 6  Annual distribution of articles per topic.**

various topics until 2020, followed by fluctuations in subsequent years, with growing interest in all the main research topics represented.

Figure 7 shows the distribution of articles by specific topic within four macro-categories: conformance checking, process discovery, process improvement, and support areas. Each column of the histogram represents the total number of articles for a given specific topic. The graph highlights that process discovery algorithms and process mining applications are particularly prominent in recent research, suggesting a trend toward process optimization through advanced techniques and their practical implementation. Despite the diversity of topics covered, these two specific areas emerge as the most studied, indicating a strong interest in the research community to improve and apply process mining methods.

We also evaluated the distribution over the years, starting from 2014, of the documents published for each process mining topic. Figure 8 shows a bar graph for each topic
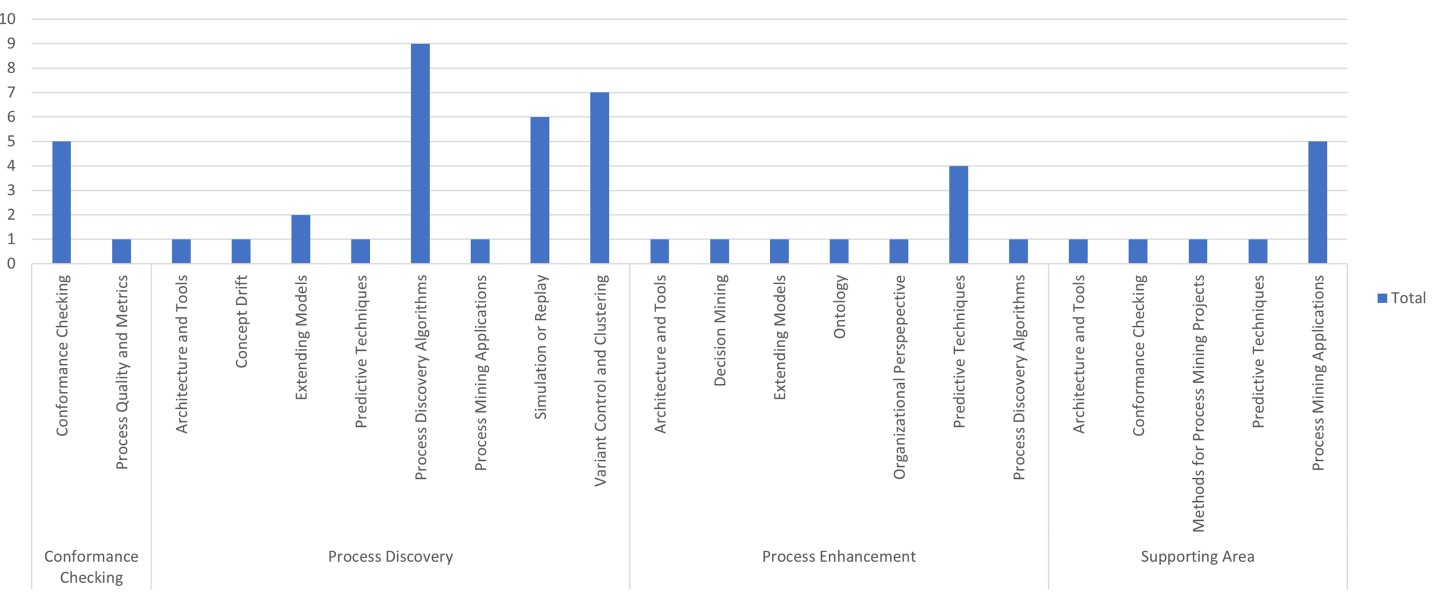

**Figure 7  Distribution of specific topics by main topic.**

**Figure 8  Temporal distribution of documents by topic.**

considered, reporting at the top left (a) the published documents relating to conformance checking; at the top right (b) those relating to process discovery; at the bottom left (c) those relating to process enhancement and finally at the bottom right (d) the published documents relating to the support areas. From the figure, we can see that in the first graph, entitled "Conformance Checking", a significant peak is observed in 2020, with a value above 3.5. In other years, such as 2014, 2021, and 2023, there are lower values, suggesting a less constant interest in this aspect, with a clear surge in 2020. The second graph, "Process Discovery", shows a more distributed trend over time. From 2016 to 2021, there is a constant growth with two peaks, one in 2019 and another more marked one in 2021, almost reaching a value of 8. 2023 and 2024 maintain a high level of interest, indicating continuity in this area of study. In the third graph, "Process Enhancement", the focus is between 2017 and 2020. Peaks are observed in 2018 and 2020, with values exceeding 4, while in other years the values are generally lower. This suggests a concentration of studies or activities during these periods, followed by a decline in 2021 and a slight increase in 2023 and 2024. Finally, the "Supporting Areas" graph shows a more irregular distribution with a first peak in 2015, then a decrease, and subsequently a new increase in 2019 and 2021. From 2021 to 2024 the values remain relatively stable, around 1.5-2, indicating continued but not growing interest in these support areas. In summary, the graphs show different trends in the various sectors analyzed, with some peak periods highlighting greater interest or specific activity. The representation suggests distinct temporal trends for each area, reflecting the changing dynamics and priorities in the studies and activities conducted over the years.

Finally, to complete the analysis relating to the first research question we also analyzed whether there was a particular type of disease most covered in the process mining documents considered. Figure 9 shows a bar graph representing both the number (in blue) and the percentage (in yellow) of published articles relating to specific diseases. Each blue bar indicates the absolute count of articles for a given disease, while the corresponding yellow bar shows the percentage of articles related to that disease compared to the total articles published. The X-axis shows the different diseases examined. Each disease is represented by a label under the corresponding bar. While the ordinate axis has two scales, one for the number of items and one for the percentage. The left scale indicates the total number of articles published for each disease, while the right scale indicates the percentage of articles relating to each disease compared to the total published articles. From the analysis of the graph it emerges that, for the majority of the diseases considered, the number of articles published is more or less equivalent. However, there are significant exceptions. The "Healthcare" category, which includes documents dealing with general healthcare processes, has significantly more published articles than other specific diseases. This suggests that there is greater attention in the scientific literature towards healthcare processes in general, probably because they cover a wider range of topics. Heart disease and ischemic stroke also stand out from the average with a higher number of articles published than other specific diseases. This indicates particularly intense research interest in these medical conditions, perhaps due to their prevalence and severity. In contrast, the other medical fields considered in the graph show a fairly uniform number of published articles,

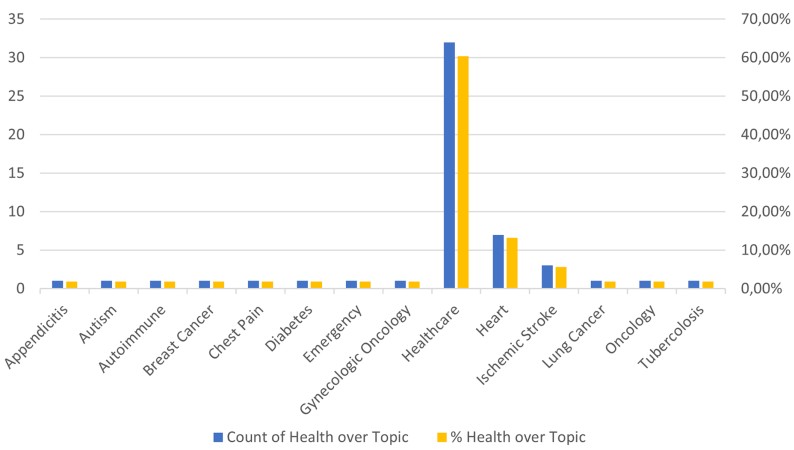

**Figure 9** Distribution of documents by specific medical field.

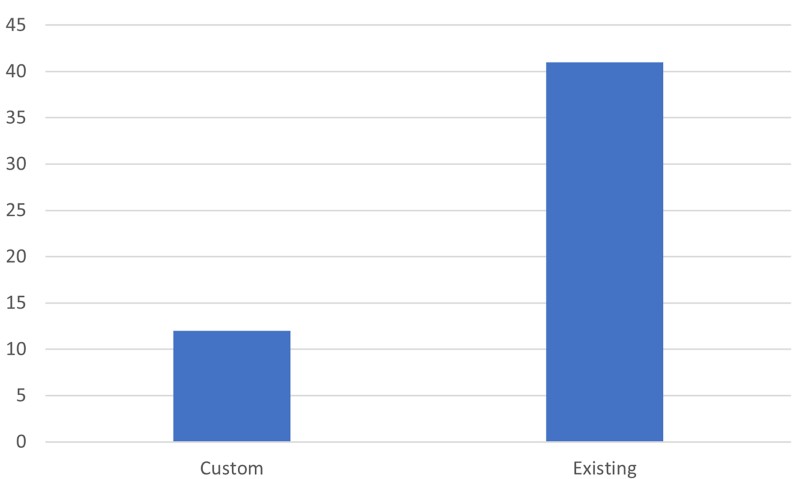

**Figure 10** Number of articles using custom techniques or already known techniques.

indicating a similar distribution of research attention among them. This balance suggests that while there are some diseases receiving increased attention, the scientific community is still dedicating significant resources to a variety of medical conditions. In summary, the graph highlights not only the general trends of research in different disease categories but also where scientific interest is most concentrated. While research on general health processes, heart disease, and ischemic stroke appears to be more prolific, other diseases continue to receive a significant amount of attention, ensuring balanced coverage in the scientific literature.

## RQ2: To what extend are process mining algorithms used in each specific healthcare application?

Understanding which process mining algorithms have been proposed and to what extent have been used in specific healthcare settings allows you to trace the field's evolution,

identify emerging trends, and recognize established techniques. This type of analysis offers a clear view of areas already explored and opportunities for future innovation. It provides useful information for professionals implementing process mining solutions in their organizations, helping them choose the algorithms best suited to their specific needs.

Figure 10 shows the number of articles using custom techniques compared to those using existing techniques. The graph indicates that a significant number of articles make use of techniques already known in the literature, with approximately 40 articles using existing techniques, compared to approximately 10 articles proposing new techniques. This suggests that, although there is innovation in the field of process mining, the majority of research is still based on established methods. To support this analysis we also specifically evaluated the techniques used in the various articles, discovering that the algorithms used are various. This variety suggests the field is dynamic and constantly evolving, with numerous techniques being explored and applied in different contexts. More specifically, the algorithms that are most used are those based on trees, support vector machine, Fuzzy Miner, and K-means.

Finally, Fig. 11 shows the distribution of algorithms in the different application fields, highlighting which algorithms are most popular and widely used in the various fields. This visualization is particularly useful for understanding the preferences of the research community, as it provides an indication of which algorithms are considered more effective or intuitive. From the analysis of the graph, a predominance of algorithms developed in the "Healthcare" sector emerges, followed by other fields, although to a lesser extent, such as "Heart" and "Diabetics". This suggests that the healthcare field in a broad sense is the one that has attracted the most attention from researchers, probably due to the relevance and practical impact that process mining can have in optimizing clinical and management processes. This aspect is important as it denotes a significant interest in the use of process analysis techniques to improve efficiency, reduce waiting times, and potentially save lives. The most used algorithms in the various fields appear to be those related to predictive techniques and process mining applications. Predictive techniques are particularly relevant in the healthcare sector because they allow us to anticipate critical events, improve diagnoses, and personalize treatments based on patient risk profiles. Process mining applications, on the other hand, allow a detailed understanding of workflows, identifying bottlenecks and inefficiencies in processes. This predominance of predictive techniques and process mining applications indicates that, despite the availability of many different algorithms, the research community tends to prefer some of them for their proven effectiveness, ease of use, or other distinctive characteristics. This trend may derive from the fact that such algorithms are more easily implementable in healthcare facilities or that they offer greater interpretability of the results, a fundamental aspect in a context where decisions must be supported by clear and understandable evidence. Furthermore, the presence of algorithms in other sectors such as "Heart" and "Diabetics" could reflect specific clinical needs in these areas, where process analysis can support the monitoring of chronic conditions or the identification of optimal care pathways.

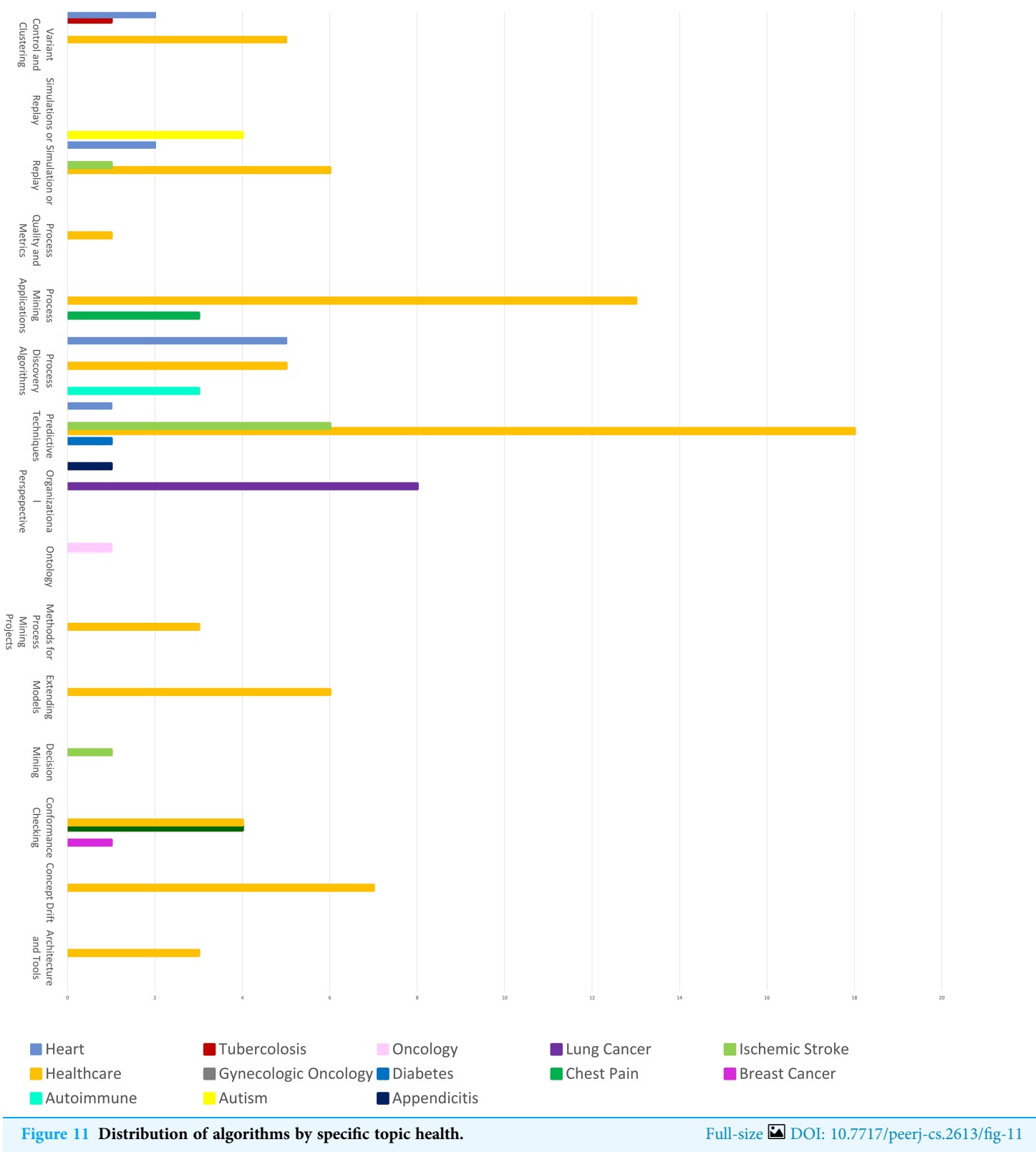

**Figure 11 Distribution of algorithms by specific topic health.**

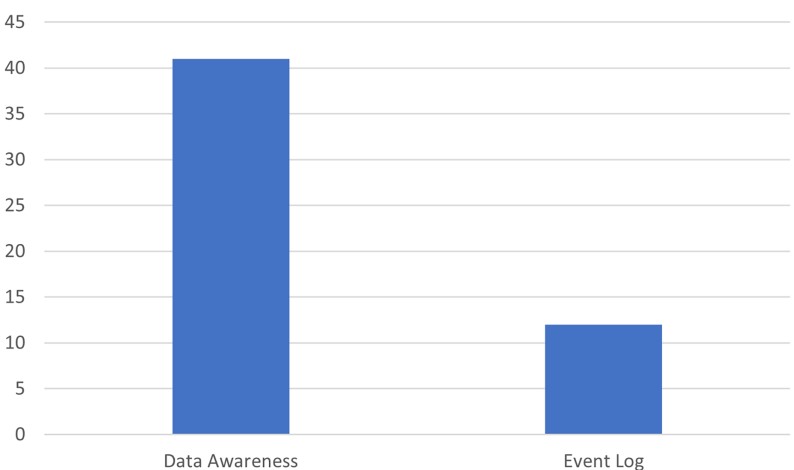

**Figure 12 Distribution of documents by data types used.**

## RQ3: What are the types of data and their characteristics?

Knowing the types of data used in process mining analyses is essential to evaluate the quality and relevance of the results because the data constitute the basis on which the analyses are built and, therefore, understanding their nature affects the validity of the conclusions.

Figure 12 clearly shows two categories of data: "Data Awareness" and "Event Log". The bar graph shows the categories on the x-axis and the number of documents on the y-axis. It indicates that the number of articles that consider a broader range of data (Data Awareness) is significantly higher than those that exclusively consider event logs (Event Log). "Data Awareness" refers to the use of various types of data, which can include not only event logs but also other data sources such as enterprise databases, unstructured data, IoT sensors, and so on. These data are characterized by greater variety and complexity, requiring more sophisticated analysis techniques to be integrated and interpreted effectively. On the other hand, "Event Log" refers to event logs that are typically used in process mining contexts. These logs contain structured data that documents time-sequenced events, often associated with specific business processes. Event log data is generally easier to manage and analyze, as it follows a more rigid and predictable structure than other data sources.

The predominance of articles in the "Data Awareness" category suggests that research in data for process mining recognizes the importance of including a wider variety of data beyond traditional event logs. This can improve process understanding and analysis, allowing you to gain a more complete and detailed view of business operations. Therefore, the findings show that while event logs remain an important source of data for process mining, there is a growing focus on integrating various data types to best exploit the distinctive characteristics of each and improve quality and the accuracy of process mining analyses.

We also considered it interesting to understand whether the data used are exclusively public or private because the nature of the data used can influence the methodologies

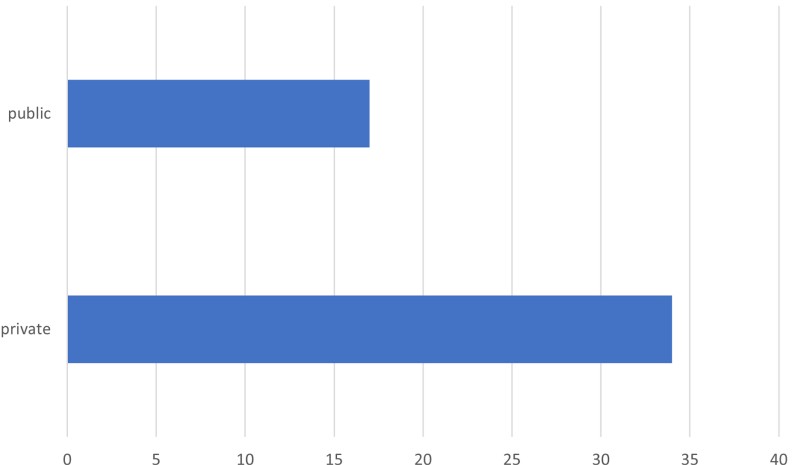

**Figure 13 Distribution of dataset types (public and private).**

adopted and the results obtained. Public and private datasets can significantly differ in structure, granularity, and complexity. Private data, for example, might offer details that are more specific and relevant to a particular organization, while public data might be more generic but less subject to access restrictions. Figure 13 shows the distribution of datasets between public and private. Most of the studies analyzed use private data, with a total of around 35 studies, while only around 20 studies make use of public data. This disparity suggests that much process mining research relies on data collected internally by organizations, likely due to the sensitive and specific nature of business processes that require protection of privacy and confidentiality. For completeness, we can also highlight that in 26% of cases, only one dataset is used, while in the remaining part, more than one dataset is used, with cases in which these reach eight or nine.

Furthermore, a qualitative analysis of privacy policies was conducted, revealing that 70.63% of the works examined did not address the concept of privacy policy, while the remaining 20.37% dealt with the issue of privacy. Only 16.67% of the works declared the adoption of data anonymization measures to protect sensitive information related to patients or other subjects, ensuring that such information is not identifiable. However, only one case explicitly stated that the use of data complies with GDPR.

Furthermore, using real data can ensure greater practical applicability of the results, while synthetic data can be useful for testing new methodologies in a controlled environment. Therefore we analyzed the presence of synthetic data in the studies considered. Figure 14 shows that most studies do not use synthetic data, while only a small part uses synthetic data. This result highlights a preference for the use of real data, perhaps to ensure greater validity and practical applicability of the results. However, the use of synthetic data may be relevant in contexts where the availability of real data is limited or where particularly stringent privacy restrictions exist. Table 3 combines these two dimensions, considering the use of synthetic data and the public or private nature of the datasets. Of the 53 studies analyzed, six use synthetic private data, while four use synthetic public data. The majority of studies, however, do not use synthetic data, with 28 studies

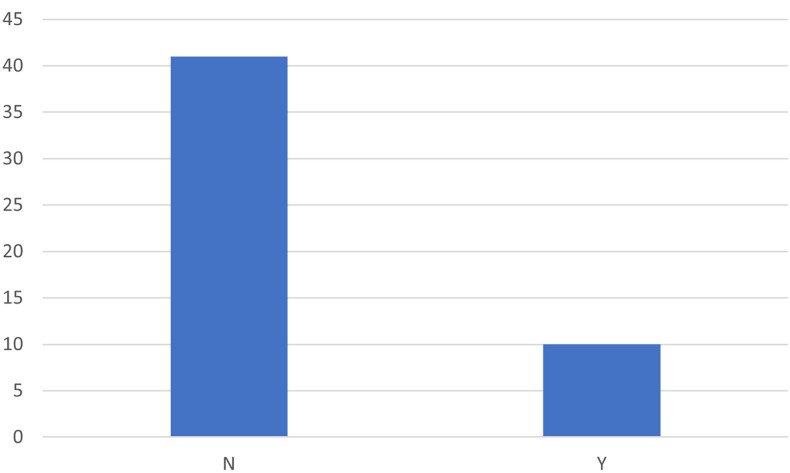

**Figure 14 Distribution of articles based on the use of synthetic data.**

**Table 3 Contiguity table on the use of synthetic data.**

Table of contiguity

|  | Syntethic | |
|  | Yes | No |
|---|---|---|
| Public | 6 | 28 |
| Private | 4 | 13 |
| Total | 53 | |

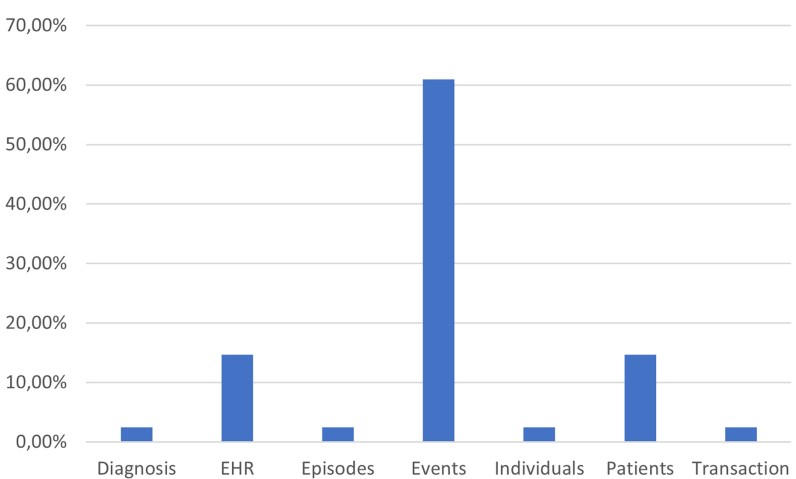

**Figure 15 Attributes used as 'case ID' and their relative percentage across articles.**

using real private data and 13 studies using real public data. This pattern further confirms the predominance of the use of real data, both public and private, underlining the importance of data veracity in process mining analyses.

Finally, we also considered it important to understand the characteristics of the data, such as the presence of specific attributes considered "case ID" (*e.g.*, events, patients, episodes), because it is crucial for determining the granularity and traceability of the analyses. This allows us to understand how processes are modeled and which aspects are most emphasized in process mining studies. In this regard, Fig. 15 shows the various characteristics considered as "case ID" in the different articles. The most common category is "Events" with a percentage above 60%, followed by "Patients" and "Episodes" with lower percentages. Other attributes such as "Diagnosis", "EHR" (Electronic Health Records), "Individuals" and "Transaction" are considered to a much lesser extent. This result reflects process mining's emphasis on event traceability, which is essential for detailed process analysis.

# DISCUSSION

This section reports some implications highlighted from the results of the three research questions and outlines future research issues on process mining in the healthcare context. This systematic review allows us to identify key research topics and areas of innovation while highlighting gaps and open challenges to be addressed. In particular, from the results of RQ1, it emerges that workflow optimization and process discovery are the main issues investigated by the scientific community. The algorithms reviewed in RQ2 highlight a wide range of approaches for analyzing and improving healthcare processes. At the same time, RQ3 reports detailed information on the data types processed, pointing out the specific need to adapt process mining tools to clinical data.

In particular, the quantitative analysis for RQ1 shows that the most studied topic is process discovery, indicating a shared interest in identifying and simulating healthcare processes from data. This emphasis reflects the growing dedication to gaining in-depth knowledge of contemporary clinical workflows in order to facilitate better decision-making and increase operational effectiveness. Although conformance checking is not discussed much, this indicates a possible area of development that could be particularly useful in ensuring that healthcare procedures adhere to legal requirements, which is critical in a regulated industry such as healthcare. Furthermore, the predominance of the use of existing techniques, rather than the development of new methodologies, suggests that the field of process mining in healthcare is still in a consolidation phase, with a priority on practical application rather than the creation of new techniques. The implications of these findings suggest the need for specific guidelines for the application of process mining in the healthcare context. Since process discovery seems to be the dominant technique, researchers and practitioners must focus on careful data collection and preparation, as data quality has a direct impact on the effectiveness of discovery techniques.

By creating algorithms specifically designed for the healthcare industry, compliance monitoring gaps may be closed and process compliance with healthcare standards might be better supported. The findings show that there is potential to increase research on new applications and tailored algorithms, even though the majority of the study is concentrated on process discovery and particular tools like Disco and ProM. Clinical professionals can make better decisions and maximize the efficiency of process mining solutions in

enhancing clinical workflows and resource management by being aware of the distinctions between the various approaches that are available and the implementation requirements linked to the different tools. Thus, a more dynamic ecosystem with tools and algorithms tailored to the particular needs of the healthcare industry can be fostered by this emphasis on innovation.

For RQ2, the analysis confirms that, although some new techniques have been introduced, most studies rely on well-established algorithms, such as decision trees, support vector machines, fuzzy miners, and K-means. This indicates that rather than focusing on methodological innovation, healthcare process mining research is still mainly focused on practical application. Current algorithms prove reliable for many applications, proving to be adaptable and suitable for handling the complexity and heterogeneity of healthcare data. However, the presence of customized methods suggests a growing interest in adapting algorithms to meet the specific needs of the clinical context, a trend that could stimulate the creation of additional specialized methods in the future. This phenomenon is particularly relevant since it implies an evolution towards more tailored solutions, which could ensure greater precision and relevance in the analyses applied to healthcare data.

For professionals who want to implement process mining solutions in their healthcare organizations, these results offer useful guidance. On the one hand, the predominance of established techniques highlights that commonly used algorithms are considered reliable and have been extensively tested, which can facilitate their integration into existing workflows. On the other hand, the variety of techniques used and the interest in developing tailored methods indicate that the field is dynamic and constantly evolving, opening up opportunities for further customization to meet specific clinical needs.

The results emerging from the distribution of algorithms across healthcare sectors show that the distribution has some important implications. First, the focus on general healthcare indicates a strong demand for tools that can improve clinical workflows, optimize resource management, and increase process efficiency, while reducing patient waiting times. Second, the significant presence of algorithms in specific domains such as "Heart" and "Diabetics" reflects the clinical needs of these sectors, where monitoring chronic conditions and personalizing treatments require a detailed analysis of care pathways and resources employed. This concentration of algorithms in key sectors also highlights the need to develop algorithms and methodologies that can respond to specific clinical requirements. For example, Predictive Techniques are particularly useful for anticipating critical clinical events and personalizing treatments, while Process Mining Applications provide an in-depth view of workflows, identifying bottlenecks and inefficiencies. These techniques not only support more efficient management of clinical processes but can also foster greater transparency in care pathways, improving communication between healthcare professionals and patients. Finally, the implications of this analysis extend to future lines of research: although the "Healthcare" sector is well explored, there are still specific areas that could benefit from further investigation. For example, areas such as emergency management or applications related to oncological pathologies could benefit from further studies that implement more personalized process mining algorithms suited to handle the peculiarities of these areas. For healthcare

professionals, this analysis represents a useful guide for choosing the algorithms best suited to their specific needs, while for researchers it constitutes an incentive to explore new areas of application of process mining, contributing to innovation and advancement of the sector.

Finally, the analysis for RQ3 reveals a definite domination of the "Data Awareness" category, which encompasses a range of sources outside of conventional event logs, including corporate databases, unstructured data, and Internet of Things sensors. This diversity shows that the discipline is evolving, with an increasing emphasis on the necessity of integrating various data to obtain a more comprehensive understanding of therapeutic processes. This multifaceted strategy is especially novel for the healthcare industry, where more sophisticated analysis is needed due to the intricacy of workflows and the variety of data sources. The findings suggest that practitioners should carefully consider the nature and source of data, as they impact the quality and applicability of analyses. The prevalent use of private data, combined with a preference for real data over synthetic data, reflects a tendency to anchor analyses to specific contexts and to ensure greater veracity and practical applicability of results. Healthcare organizations intending to adopt process mining should therefore invest in access to detailed, real data and consider integrating multiple data sources to enrich analyses. For the research community, these findings highlight the need to balance the use of real and synthetic data. Synthetic data, although less used, can play an important role in testing new methodologies and addressing access limitations due to privacy concerns. Furthermore, the distinction between public and private data suggests an opportunity to create more detailed and specific public datasets for process mining, which could facilitate the reproducibility of studies and stimulate further innovation. The predominance of the "Events" attribute as case ID in the data suggests a strong focus on temporal traceability, which is crucial for accurate process analyses. However, the inclusion of other attributes, such as "Patients" and "Episodes," opens up new possibilities for a more focused analysis of specific care dynamics and treatment pathways. An approach incorporating different granularity attributes could further improve the understanding of clinical processes, making process mining analyses more adaptable to a broad spectrum of applications in the healthcare sector. Therefore, a greater awareness of data characteristics and their strategic selection can improve the effectiveness of process mining analyses, offering concrete perspectives for the optimization of clinical workflows and the management of healthcare resources. Finally, The results of the qualitative analysis of privacy policy enforcement suggest an important gap in the consideration of privacy policies within healthcare process mining. Although anonymization is recognized as a crucial measure to protect data confidentiality, its application is still limited, which may hinder the wider adoption of process mining in clinical settings where data protection is paramount. The lack of explicit compliance with GDPR in almost all the analyzed studies highlights the need for greater attention to privacy regulation, which is particularly relevant in healthcare. For healthcare organizations and researchers, these results indicate the importance of integrating security and privacy measures into their process mining projects. The adoption of anonymization policies and compliance with regulations, such as GDPR, not only strengthen the protection of patient

data but also facilitate greater acceptance and trust in the use of these technologies. For the research community, this represents an opportunity to develop and promote guidelines that ensure data protection, helping to create a safe and compliant environment necessary for the diffusion and application of process mining in healthcare.

## CONCLUSIONS

This study delves into the field of process mining in healthcare by conducting a systematic review. We begin by providing an overview of process mining and a detailed classification of process mining topics covered in the literature.

Next, we describe the research methodology used, beginning with the development of specific research questions to guide the review process. Next, we describe the application of inclusion and exclusion criteria to refine the selection of relevant studies, culminating in a final set of 100 articles for in-depth analysis.

The literature classification includes several dimensions, including relevant topics, specific techniques and algorithms used, and types of data used. In particular, this last aspect, *i.e.*, data types, has not been addressed in previous reviews.

Furthermore, we examine the evolution of research trends over time within the above-mentioned review dimensions.

In conclusion, this review builds on existing knowledge by updating, confirming, extending, and improving information from previous literature reviews on process mining in healthcare. Our analysis demonstrates that process mining in healthcare is a rapidly expanding field of research, with a growing body of literature spanning a wide range of topics.

### Funding
The authors received no funding for this work.

### Competing Interests
Martina Iammarino and Lerina Aversano are Academic Editor for PeerJ.

### Author Contributions
- Lerina Aversano conceived and designed the experiments, analyzed the data, authored or reviewed drafts of the article, and approved the final draft.
- Martina Iammarino performed the experiments, analyzed the data, authored or reviewed drafts of the article, and approved the final draft.
- Antonella Madau performed the computation work, prepared figures and/or tables, and approved the final draft.
- Giuseppe Pirlo conceived and designed the experiments, authored or reviewed drafts of the article, and approved the final draft.
- Gianfranco Semeraro performed the experiments, performed the computation work, prepared figures and/or tables, and approved the final draft.

## Data Availability

This is a literature review.

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
