# Peer review of "Process mining applications in healthcare: a systematic literature review"

_PeerJ Computer Science, doi:10.7717/peerj-cs.2613_

## Round 0.1 · original submission · Major Revisions

Based on the external reviews, a major revision is needed for further consideration. See the reviews for more information.

Reviewer 1 ·

Basic reporting

The manuscript entitled “Process Mining techniques and applications in healthcare: a Systematic Review” has been investigated in detail. The manuscript addresses an important topic—process mining in healthcare—by reviewing existing literature and highlighting algorithms, themes, and challenges. However, the lack of novelty, superficial treatment of key topics, and absence of rigorous quantitative analysis weaken its impact. Additionally, the review methodology lacks clarity, and the practical guidelines promised in the abstract are not delivered effectively. A major revision is necessary to strengthen the contribution of this paper, including deeper analysis of algorithms, challenges, and more focused practical recommendations. Without significant improvements, the manuscript risks being redundant with existing reviews in the field.
1) The manuscript presents a systematic review of process mining in healthcare but lacks novelty. Many of the observations, such as the potential of process mining to optimize workflows and reduce operational costs, are already well-established in the literature. The paper does not present new findings, insights, or methodologies, which is essential for making a meaningful contribution to the field.
2) While the paper claims to provide an overview of the algorithms used in healthcare process mining, the discussion remains superficial. The manuscript lacks in-depth technical explanations and comparisons of these algorithms. It should discuss why certain algorithms are preferred in specific healthcare scenarios, their limitations, and how they address challenges unique to healthcare data.

Experimental design

3) The manuscript briefly mentions challenges related to healthcare data management, but this section is underdeveloped. Important challenges like data privacy concerns, integration of heterogeneous data sources, and handling of missing or incomplete clinical data should be addressed in detail. Furthermore, the paper should explore how process mining can be applied while adhering to strict regulatory requirements like HIPAA and GDPR.
4) The methodology for selecting and analyzing the articles is not sufficiently transparent. The paper does not clarify the criteria for inclusion or exclusion of studies in the review process. A detailed explanation of the search strategy, databases used, and how the final set of articles was selected is essential to ensure the rigor of the systematic review.

Validity of the findings

5) The paper would benefit from some form of quantitative analysis, such as metrics showing the frequency of different algorithms used, the prevalence of process mining applications in various healthcare sectors, or a breakdown of data types by algorithm. This would provide more concrete evidence of trends in the field and add rigor to the review.
6) The manuscript makes broad claims about the potential benefits of process mining without sufficient empirical evidence or case studies to support these claims. For example, it is stated that process mining improves operational efficiency and clinical workflows, but the manuscript does not cite specific examples or provide quantitative results from real-world applications. Including case studies or more detailed examples from the reviewed literature would strengthen the paper.

Cite this review as

Reviewer 2 ·

Basic reporting

The article aims to systematically review the literature on the use of process mining in healthcare. However, the objectives are not clearly stated, and the scope of the review lacks focus.

There are variations in methodologies, data sources, and healthcare. These aspects are important and should be considered. The article did not described challenges in applying process mining in healthcare, such as data privacy and workflow complexity.

There are some formatting issues. For example, references on line 51 and 71. Moreover, line 103 has ??.
Authors should revise carefully the manuscript.

Experimental design

The article mentions that specific criteria were used to include or exclude studies, but the rationale behind these criteria is not well explained. The search strategy used to identify relevant studies is inadequately described. Authors mentioned some databases (e.g., Science Direct, IEEE Xplore), the keywords used, and the timeframe of the search. The article does not clearly define how data from the selected studies were extracted or analyzed.

Validity of the findings

This article presents an overview of various applications of process mining in healthcare, but it is not well-structured. The discussion of different techniques and their benefits is not well explained and lacks critical analysis. Authors should provide more detailed explanations and examples of how these techniques are applied in specific healthcare scenarios (e.g., patient flow optimization, resource management) would strengthen the article. Moreover, there is limited detail of quantitative outcomes from the reviewed studies.

Cite this review as

---

## Round 0.2 · accepted · Accept

Based on the feedbacks of external reviews, the manuscript is now ready for acceptance. The survey paper will benefit the fields of process mining applications in healthcare.

Reviewer 1 ·

Basic reporting

My comments have been addressed. It is acceptable in the present form.

Experimental design

My comments have been addressed. It is acceptable in the present form.

Validity of the findings

My comments have been addressed. It is acceptable in the present form.

Cite this review as